# Impact of Autoclaving on the Material Properties of Vat-Photopolymerization-Produced Components Intended for Bioprocess Engineering

**DOI:** 10.3390/ma18204720

**Published:** 2025-10-15

**Authors:** Lauri Hoffmann, Bruno Gallace, Clara Herr, Kai Scherer, Adrian Huwer, Percy Kampeis, Roland Ulber, Michael Wahl

**Affiliations:** 1Institute for Operations and Technology Management, Trier University of Applied Sciences, Environmental Campus Birkenfeld, Campusallee, 55768 Hoppstädten-Weiersbach, Germany; lauri.hoffmann@umwelt-campus.de (L.H.); b.gallace@umwelt-campus.de (B.G.); k.scherer@umwelt-campus.de (K.S.); a.huwer@umwelt-campus.de (A.H.); 2Institute of Biotechnical Process Design, Trier University of Applied Sciences, Environmental Campus Birkenfeld, Campusallee, 55768 Hoppstädten-Weiersbach, Germany; clhr2771@umwelt-campus.de (C.H.); p.kampeis@umwelt-campus.de (P.K.); 3Institute of Bioprocess Engineering, University of Kaiserslautern-Landau, Gottlieb-Daimler-Straße 49, 67663 Kaiserslautern, Germany; roland.ulber@mv.rptu.de

**Keywords:** additive manufacturing, stereolithography, steam sterilization, mechanical properties, surface properties, autoclavable, vat photopolymerization

## Abstract

Due to a lack of investigated materials for the additive manufacturing of multi-use functional parts in bioprocess engineering, this study aimed to evaluate the influence of multiple autoclaving cycles on the properties of a heat-resistant material (xPeek147) printed with vat photopolymerization. Sample bodies were tested regarding their mechanical properties of tensile strength, elongation at break, and Charpy impact, as well as surface properties of roughness and wettability after up to 50 autoclaving cycles (121 °C, 2 bars, 15 min). The tightness was checked after up to 20 cycles, and accuracy was inspected for manufactured benchmark bodies after up to 10 autoclaving cycles. The reported results showed no significant changes in tensile strength, elongation at break and Charpy impact after 20 cycles, but a significant decrease after 50 autoclaving cycles, accompanied by microcracks in the structure. Regarding the surface properties the material retained its hydrophilicity, and the surface roughness was not affected significantly. No changes in tightness occurred, and the benchmark bodies for dimensional changes showed no process-relevant deviations. Through the investigations, a material for the additive manufacturing of multi-use functional parts for bioprocess engineering was identified. Additionally, a testing method for materials with the same intended application was provided.

## 1. Introduction

Additive manufacturing (AM) offers a great range of methods, materials, and applications. Possible applications vary from large-scale printed buildings made of concrete to the printing of small-scale medical implants or tissues [1,2]. In recent years 3D-printing technologies have also been used in the field of biotechnology. There AM enables the specified manufacturing of microfluidic devices, the bioprinting of living cells, and the supply of functional parts for bioprocess engineering or specified labware [3,4,5]. In previous studies, for instance, AM was used for optimized geometries of stirrers or supports for fungi sporulation in agitated vessels [6,7]. Further studies printed bioreactor vessels with integrated features for sparging and harvesting in submerse cultivation [4]. Other approaches manufactured parts of a photobioreactor prototype and biocarriers for the emerse cultivation of cyanobacteria [8,9].

For bioprocess engineers, the two requirements of sufficient part accuracy and heat stability are important criteria when choosing an appropriate additive-manufacturing technology and material. Heat stability is required for the integrity of the parts in case of heat sterilization like autoclaving (e.g., 121 °C, 2 bars, and 15 min duration) as a commonly used sterilization procedure in bioprocess applications [3]. Heat stability is a challenge, especially for polymer parts made by AM. Therefore, previous studies investigated the influence of different sterilization methods on material properties. The results show a high variability due to the differences in the materials and the AM methods used. The reported effects regarding the mechanical properties vary between an improvement in tensile strength and failed tensile tests due to cracks in the specimens after autoclaving procedures [10,11,12].

Regarding the required accuracy of autoclaved AM parts, studies reported various effects. In the study by Told et al., they detected a range of deformations [10]; these varied from complete deformation for materials like polylactic acid (PLA), acrylonitrile butadiene styrene (ABS), or polyethylene terephthalate (PETG) to no deformations or a slight increase in the specimens’ cross-section depending on the used material and the AM process. In an application-oriented approach, Rynio et al. reported deformations after autoclaving for aortic stent grafts for surgical-use printed materials like PLA, PETG, or Polypropylene (PP) in material-extrusion processes. Contrary to that, materials used in vat-photopolymerization AM of Formlabs (Somerville, MA, USA) stayed stable [13]. In vat photopolymerization in general, polymer resins are hardened layer by layer through photopolymerization [14]. Other studies using this process show variability in dimensional changes and position accuracy by autoclaving, ranging from no significant changes to deformations depending on the combination of resin material and vat photopolymerization machine [15,16,17,18]. Therefore, for each combination, a specific investigation is necessary to understand the deformation behavior caused by autoclaving.

Other main requirements for bioprocess engineering concern the tightness and the surface properties of AM parts. Tightness is a challenge, especially for parts manufactured by material-extrusion processes [19,20,21,22]. Printing with vat photopolymerization offers the possibility to produce tight parts. The tightness of such parts was validated in previous studies [23,24,25], but, to our knowledge, was not checked after sterilization procedures.

Surface properties like roughness or wettability influence the adhesion behavior of microorganisms on surfaces [9,26,27,28]. The surface properties of AM parts, for instance, made by vat photopolymerization, depend on the used printing parameters and bulk material properties [29]. Regarding the influence of sterilization on surface properties, the study by Told et al. examined the effect of different sterilization protocols on the surfaces of AM polymer parts by electron microscopy. For autoclaved parts produced by stereolithography, no changes were detected in the visible structure [10]; however, for the determination of changes in wettability or roughness, measurements are necessary.

Some of the properties of AM parts have been investigated in previous studies to determine whether they are affected through autoclaving or other sterilization procedures. Most of them checked only one autoclaving cycle, which results in a suitability just for single-use applications in medical contexts [11,16,30,31].

Only a few studies treated their specimens produced by vat photopolymerization with more autoclaving cycles. In the study by Gensler et al., they looked for changes in the mechanical properties after three autoclaving cycles and detected no significant change [25]. The study by Told et al. [10] expanded the investigations to 10 and 20 autoclaving cycles, with the intention of multiple uses in medical devices. Through autoclaving, vat-photopolymerization-processed material specimens changed in size, and fissures and cracks occurred at the surface of the tensile specimens. Consequently, a complete assessment of the mechanical properties was not possible. They recommended this material just for applications that have no requirements for accuracy or structural integrity [10].

Nevertheless, vat photopolymerization offers potential for the additive manufacturing of structural parts for bioprocess engineering, overcoming the mentioned disadvantages of material-extrusion processes and the materials used in them. The vat-photopolymerization process allows for printing tight parts with better surface qualities and higher precision than other printing technologies [23,24,32,33]. Additionally, vat printers, which cure the whole cross-section layer of the part with a single illuminated picture projection, offer fast processing times [32]. Furthermore, the range of possible resin materials enables specified material selection if individual mechanical or thermal properties are required [33].

Consequently, the previous approaches did not identify a possible vat-photopolymerization resin material that retains its properties across multiple autoclaving cycles and, therefore, is suited for multi-use application in bioprocess engineering. Especially in bioprocess applications, AM components are often used as single-use items or prototypes [34,35]. However, due to increasing sustainability demands, enabling multiple-use cycles has become more essential. This requires a deeper understanding of the limitations of AM components under heat sterilization processes such as autoclaving. Improving their durability can extend their service life and significantly reduce costs, making AM a more sustainable and economical option.

Therefore, the novelty of the approach described in this paper is the evaluation of the influence of up to 50 autoclaving cycles on the material properties of a potential material. As criteria for a comprehensive evaluation, the mentioned mechanical and surface properties, as well as the tightness and geometrical accuracy, were tested after defined numbers of autoclaving cycles. Afterwards, they were compared to those of reference specimens. Besides the testing of a potential material, it was an additional intention to provide a methodology of material testing for AM materials with intended use in bioprocess engineering. According to the general character of this study, no investigations were performed regarding cytotoxicity and biocontamination susceptibility, as this depends on the intended bioprocess and the microorganism used.

## 2. Materials and Methods

### 2.1. Additive Manufacturing of Specimens

Specimens of all kinds were produced using the vat-photopolymerization technology according to DIN EN ISO/ASTM 52900 [14]. The 3D-printer NEXA XiP Pro (NEXA3D Inc., Ventura, CA, USA) was used, which cures the resin with a liquid-crystal display (LCD) and a pixel size of 46 µm [36]. As a temperature-resistant and mechanically resistant material, xPEEK147 HDT230 High Heat (Henkel AG & Co. KGaA, Düsseldorf, Germany) was chosen [37]. Before printing, the resin was warmed to 35 °C for three hours. For post-processing, the parts were cleaned in isopropyl alcohol twice for two minutes, respectively. After drying for 60 min, the specimens were cured twice in a UV-light chamber for 30 min, flipping them over after one cycle. Afterwards, they were baked for three hours at 170 °C according to manufacturer recommendations [38]. Before holding a constant temperature, they were warmed up with an increment of 1.5 °C per minute. After baking, the specimens were cooled down to room temperature in the oven. All material test specimens were printed with a layer height of 100 µm, standing upright on the smallest side at an angle of 80° to the building platform, inclined over the smallest edge.

### 2.2. Autoclaving Method

All sterilized specimens were autoclaved by moist heat in a standard autoclave Systec VX/VE Series (Systec GmbH Labor-Systemtechnik, Wettenberg, Germany). The autoclaving routine used a temperature of 121 °C at an absolute pressure of 2 bars for 15 min. During the autoclaving routine, the specimens were treated with a maximum heating rate of 1.19 °C per minute, and cooled down at a maximum rate of 0.68 °C per minute. The specimens were stored in pots of water during and after autoclaving.

### 2.3. Testing Methods

To investigate the influence of autoclaving in multiple cycles on material properties, static mechanical tests were applied. Separated benchmark bodies were used to determine potential geometrical changes through multiple autoclaving cycles. Parts with specialized geometry were printed for the tightness test. Results between specimen batches were tested afterwards for their statistical significance (*p* < 0.05). A summary of all specimen types and their treatments is presented in Table 1. In this document, the different autoclaving batches (AU) were labeled depending on their number of autoclaving cycles (n) and abbreviated into AU (n). For example, the specimens in batch AU 10 were autoclaved ten times.

A total of 60 specimens for tensile and Charpy tests, respectively, were printed. Additionally, two benchmark bodies and five tightness-test bodies were manufactured.

#### 2.3.1. Tensile Testing

For tensile testing, a Shimadzu Autograph AGS-X Series (Shimadzu Deutschland GmbH, Duisburg, Germany) with a 10 kN load cell and without an extensometer was used. The samples were designed with the dimensions shown in Figure 1. The formatting of the drawing is based on DIN EN ISO 527-2:2012-06 [39]. DIN EN ISO 527-1:2019-12 [40] was selected as the method for the test procedure. The tensile strength and the elongation at break were determined. For each autoclaving batch, ten specimens were tested.

#### 2.3.2. Charpy Impact Test

To determine Charpy impact strength, a Zwick Roell 5102.202 test apparatus (Zwick GmbH & Co., Ulm, Germany) was used. The test was carried out with a 0.5 J pendulum hitting the narrow side of the specimen. The parts had dimensions of 4 mm × 10 mm × 80 mm without a notch (type: ISO 179-1/1eU), according to DIN EN ISO 179-1:2023-10 [41]. Ten samples per batch were inspected.

#### 2.3.3. Roughness Testing

Roughness tests were carried out with a tactile roughness measurement instrument, Mahr Perthometer PGK (Mahr GmbH, Göttingen, Germany), and evaluated with software MarWinEasyRoughness Version v11.20-03 SP1 (Mahr GmbH, Göttingen, Germany). Values of arithmetically averaged roughness (R_a_) were determined at the large sample side of ten Charpy-impact-test samples for each batch. Additionally, the specimens were investigated for visible changes in the surface with a Zeiss Stemi 305 microscope (Carl-Zeiss Microscopy GmbH, Jena, Germany) and an added MicroCam SP 5.0 digital camera (Bresser GmbH, Rhede, Germany).

#### 2.3.4. Surface Energy Determination

For surface energy investigations, contact angle measurements employing a DataPhysics OCA 15 plus with SCA20 software (DataPhysics Instruments GmbH, Filderstadt, Germany) were carried out. For testing, three drops, 4 µL each, per fluid were placed with a dosing rate of 10 µL/s on the large side of the Charpy impact test samples. As fluids, water and diethylene glycol were used. The test was repeated for five specimens per autoclaving batch. Values of surface energy were calculated software-based with the Owens-Wendt and Kaelble method, which is described, for example, in DIN EN ISO 19403-1 [42,43,44].

#### 2.3.5. Scanning Electron Microscopy

To investigate the relationships between mechanical properties and structure, Scanning Electron Microscopy (SEM) investigations of the samples’ surface were performed using a JEOL JSM-6610 (Jeol Ltd., Tokyo, Japan). An SEM Coating Unit E5100 (Polaron Equipment Ltd., Watford, Hertfordshire, UK) was used for gold coating of the samples. SEM analysis was performed on two randomly selected specimens from all autoclaving batches after the Charpy impact test.

#### 2.3.6. Tightness Testing

Special bodies with air-pressure-tight connectors were designed to check the influence of autoclaving on the tightness properties of the material. A detailed geometry sketch is presented in Figure 2. These parts were loaded with 0.5 bars of overpressure to the environment on the inside. For a qualitative check, the parts were immersed in water and checked for bubbles. Previous studies of printing-quality assessment with a comparable method were performed by Gordeev et al. [19].

#### 2.3.7. Testing for Dimensional Changes

As benchmark bodies for testing dimensional changes over the autoclaving cycles, a simple three-blade stirrer with 30° angled blades was chosen. For checking geometrical changes across the autoclaving cycles, the surfaces of the parts were recorded through 3D scanning with a Zeiss Comet L3D 5M (Carl-Zeiss Optotechnik GmbH, Neubeuern, Germany), and a virtual surface comparison was carried out with Inspect Plus software version 5.53.376 (Carl-Zeiss Optotechnik GmbH, Neubeuern, Germany). The specimens were sprayed with a matting coating before recording and were cleaned afterwards. For evaluation of dimensional changes, the recorded surface mesh data were positioned by best-fit alignment to their corresponding CAD-model data. Afterwards, a color plot was evaluated regarding deviations in form and dimension. Two stirrers were printed and scanned after printing, and after five and ten autoclaving cycles, respectively.

### 2.4. Statistics

The results of the material investigations were examined for normality by the Shapiro–Wilk test and for homogeneity of variances by Levene’s test.

If these assumptions of Analysis of Variance (ANOVA) were not met, a Kruskal–Wallis test was carried out. Significant differences between charges were identified by post hoc Dunn’s test with Bonferroni’s correction. Results were called significant for *p* < 0.05 [30]. As software for analysis and graphic design, Origin Version 2024b (OriginLab Corporation, Northampton, MA, USA) and R Version 4.4.1 [46] were used.

## 3. Results

### 3.1. Tensile Testing

The mean results of tensile testing ranged from 39.47 MPa (AU0) to 20.65 MPa (AU50). The elongation at break reached values of 1.53% (AU15) to 0.75% (AU50). All batches have higher mean values than AU50. Due to failed normal distribution, a Kruskal–Wallis test was carried out, which pointed to differences for ultimate tensile strength (*p* < 0.001) and elongation at break (*p* = 0.001). For both instances, a significant decrease was detected for AU50 compared to AU0, AU5, and AU15 by the post hoc test. The results are summarized in Table 2 and visualized by boxplot in Figure 3. According to DIN EN ISO 527-1:2019-12 evaluations of the tension–elongation behavior, the material specimens display brittle material properties [40]. At least five specimens per batch were tested successfully in the limits of the test method.

### 3.2. Charpy Impact Test

The results of the Charpy impact test are presented in Table 3 and visualized in Figure 4. The values range from 3.60 kJ/m^2^ (AU0) to 0.81 kJ/m^2^ (AU50). Statistical analysis through the Kruskal–Wallis test determined a significant difference between the specimen batches (*p* < 0.001). The post hoc test pointed at a significant decrease in the Charpy impact of AU50 compared to AU0 (*p* < 0.001), AU5 (*p* = 0.014), and AU15 (*p* = 0.005). For all batches except AU50, the majority of values ranged between 10% and 80% of the energy capacity of the pendulum, according to DIN EN ISO 179-1:2023-10 [41]. In the case of AU50, this requirement could not be met because the smallest pendulum was already used.

### 3.3. Roughness Testing

Mean values of arithmetically averaged roughness varied from 0.7 µm (AU0) up to 1.21 µm (AU20). Results checked with the Kruskal–Wallis test failed to display significance (*p* = 0.810); therefore, no change in surface roughness is detected across the autoclaving cycles. Table 4 lists the results of the measurement, and corresponding boxplots are provided in Figure 5.

The large deviations in roughness caused by outliers can be explained by the inhomogeneous surface of single samples; these are also visible to the eye, and depend on the position of the samples on the building platform. In the area surrounding the material tank, small crystals occurred at the surface and changed the roughness properties. For the investigations with the digital microscope, no changes caused by autoclaving were detected.

### 3.4. Surface Energy Determination

The results of determined surface energy range from 35.12 mN/m (AU0) up to 42.25 mN/m (AU20). Mean values between 78.26° and 60° were detected for the contact angle of water. A Kruskal–Wallis test was performed due to a failed Levene’s test, which revealed no significant differences between the batches (*p* = 0.247) for surface energy. After autoclaving, the mean values of the contact angles of water of each batch were lower than the reference, AU0. However, the Kruskal–Wallis test (*p* = 0.001) only showed significant differences for AU15 (*p* = 0.016) and AU20 (*p* < 0.001) compared with AU0. The comparison of batches for the contact angle of diethylene glycol failed to display significance (*p* = 0.199).

Most measured contact angles of water had values smaller than 90°. These point to hydrophilic material properties [42]. Deviations within batches are caused by outliers of single local specimen measurements, resulting in deviated calculations of the surface energy. The results are summarized in Table 5 and visualized by boxplot in Figure 6.

### 3.5. Scanning Electron Microscopy

The SEM images of AU50 showed microcracks on the surface of the specimens. Samples of the other batches (AU0, AU5, AU10, AU15, AU20) showed no comparable microcracks caused by autoclaving. Single pores caused by the manufacturing process were identified in samples of all batches. Exemplary images are provided in the Appendix A. The microcracks in the surfaces of the samples of AU50 weaken the structural integrity of the specimens and lead to the reported deteriorated mechanical properties in tensile and Charpy-impact testing, respectively.

### 3.6. Tightness Testing

No bubbles were detected during the tightness tests. As a result, all bodies are found to be tight, even after 20 autoclaving cycles. Through autoclaving, no pores occurred.

### 3.7. Testing for Dimensional Changes

For the test of dimensional changes, the two stirrers were compared with their corresponding CAD model post-processing, after five and after ten autoclaving cycles. As an example, the resulting color plot of the surface comparison of the first stirrer is presented in Figure 7. The analysis of the second stirrer showed similar results and was visualized in the Appendix A. Both stirrers showed a tendency for a slight growth regarding the outer diameters of the blades and the central shaft of the stirrer. This is visualized in Figure 7 by the increase in yellow- and red-colored surface areas at the outer diameters. A detailed determination of the local deviations in 3D distances was prepared for three positions (black squares in Figure 7). The maximum deviation was reported for the outer diameter of the blade of the first stirrer, with 0.226 mm between AU 0 and AU 10. Most of the surfaces were colored green, indicating nearly no differences.

## 4. Discussion

The purpose of this study was to evaluate the impact of up to 50 autoclaving cycles on the properties of a material manufactured by LCD-based vat-photopolymerization additive manufacturing, with a special focus on its application in bioprocess engineering. For an assessment of the results into a broader context, investigations of other materials and printing processes of other vat-photopolymerization technologies like Digital Light Processing (DLP) and stereolithography (SLA) are also considered.

With a focus on the tests for mechanical properties, the ultimate tensile strength and the elongation at break decreased between AU20 and AU50. Investigations in comparable studies reported a varying influence of autoclaving on tensile strength. Studies by Pop et al. and Valls-Esteve have shown an improvement in tensile strength for materials that are well-suited for surgical applications [11,12]. A decrease in tensile strength was described for materials intended for dental use [31], or materials without any specialization for medical use or heat compatibility [12]. According to our study, a decrease in tensile strain was also reported by Pop et al. for SLA specimens [11]. On the contrary, Linares-Alvelais et al. describe a decline in the E-Modulus through autoclaving, which leads to an increase in strain in the range of elastic material behavior [31]. It is important to emphasize that all those studies used just one autoclaving cycle.

In contrast, Gensler et al. treated their SLA-printed specimens made of Dental SG Resin specialized for dental application (Formlabs Inc., Somerville, MA, USA) with three autoclaving cycles, and detected no significant changes but a tendency towards a decrease in the tensile strength [25]. Told et al. treated white-resin SLA material (Formlabs Inc., Somerville, MA, USA) with up to 20 autoclaving cycles. In this case, tensile tests were not possible due to material failure with cracks and fissures [10]. Charpy impact tests performed by Told et al. showed no significant change across the cycles [10]. This result is in accordance with our results, where a change in Charpy properties occurred first in specimens treated by 50 autoclaving cycles.

This is in line with the SEM analysis. Charpy impact specimens of AU20 showed no microcracks in their surface. According to Told et al., SEM investigations of white SLA Charpy specimens also showed a smooth surface after the same number of autoclaving cycles [10]. In our study, deterioration of mechanical properties was first reported for the AU50 specimens, on which microcracks were detected. Comparable microcracks after one autoclaving cycle were detected by Keßler et al. for SLA-printed surgical guides [17]. Cracks, as a described symptom of polymer degradation in the literature [47], lead to damage in the sample structure and, therefore, to a decline in mechanical properties.

The reported decrease in mechanical properties occurred both for tensile and Charpy-impact-test specimens. Thus, the results indicate no relationship between the surface-to-volume ratio and the deterioration in mechanical properties.

In agreement with the other studies, our findings underlined the necessity of materials for heat compatibility. Materials for surgical use or materials with heat compatibility show constant or, in some cases, improved mechanical properties after autoclaving. Our results suggest that this only applies to a certain number of cycles. This indicates that there is a material-dependent time point at which a deterioration in mechanical properties occurs. Depending on the mechanical stress for several practical uses, such time points mark the end of life for engineered parts. In our study, the end-of-life point was reported between 20 and 50 autoclaving cycles. If a more exact individual time point is needed for a material, a detailed analysis with smaller steps (e.g., in our case: AU25, AU30, AU35, AU40, AU45, and AU50) between material investigations is necessary. Impacts of temperature and water during the autoclaving procedure cause hydrolysis and thermal-oxidative degradation [48]. We suppose that, at this time point, such effects harm the structural integrity of the material, and former material properties cannot be maintained.

In general, the reported results for tensile properties of all autoclaving batches differ from those described by the manufacturer. For non-autoclaved specimens, values of 65 MPa for the ultimate tensile strength and 2.9% for elongation at break were determined [37], compared to a maximum of 40 MPa (AU0) and 1.5% (AU0 and AU15), respectively, reported in our study. We presume that these differences are caused by different methods of pre-processing of the resin and post-processing of the specimens after printing.

Regarding the surface properties of vat-photopolymerization-printed parts, studies with comparable layer thicknesses (100 µm) reported values of Ra ranging from 0.11 µm [29] up to 3.75 µm [26] for a surface angle of 90° to the building platform, respectively. Other studies with layer thicknesses of 50 µm reached values starting at 0.224–0.252 µm (three different materials) [49], over 1.69 µm [50], up to 3.609 µm [51]. For all autoclaving batches, the results of our roughness investigations are within the range of other studies. Lee et al. described a relationship between the surface roughness and materials’ viscosity and layer thickness [29]. Milde et al. detected similar roughness for four of five tested resins with a one-machine setup. The reason for the higher roughness in the fifth was explained by a possible double curing of layers while printing due to the transparency of the resin used [52]. Therefore, the surface roughness is influenced by the printing setup and the properties of the resin. Additional research on the influence of environmental impacts on surface properties was performed by Alageel et al. with three materials for dental application [49]. As impacts, simulated brushing and thermocycling afterwards were used. They detected, for three printing angle/material combinations, a reduced Ra caused by brushing and thermocycling; however, there was no differentiation in the results between mechanical- or thermal-caused surface changes. In our study, we reported no impact caused by higher temperatures through autoclaving in up to 50 cycles.

Relatively few studies have been performed to investigate the surface energies or contact angles of vat-photopolymerization-printed and non-autoclaved surfaces. Lee et al. reported, for a comparable printing setup, a value of 79° and detected a tendency of higher contact angles for lower layer thicknesses (50 µm) and smaller building angles (0°, 45°) [29]. Yan et al. reported hydrophobic surface properties with a contact angle of 93.7° for a layer thickness of 50 µm and a building angle of 90° [50]. Previous studies by Török et al. and Told et al. investigated autoclaved surfaces by SEM. After one cycle, Török et al. reported no damage or changes in surfaces manufactured by the PolyJet printing process [53]. In line with that, the visible surface properties of the SLA-printed parts did not change after 10 and 20 cycles in the studies by Told et al. [10]. These results are consistent with our findings, as we detected no significant change in surface energy and the contact angles of diethylene glycol. The water contact angles showed hydrophilic behavior for all batches, and the significant changes between batches could not be linked to the number of autoclaving cycles.

Previous studies described the adhesion behavior of microorganisms depending on wettability properties (hydrophobicity/hydrophilicity) of the used materials, the type of microorganisms [28,54,55], and the surface roughness or the size of surface features [27,56,57]. Other authors studied the influence of printing parameters on microorganism adhesion [9,26,29]. In our study, we detected no correlation between the number of autoclaving cycles and the wettability behavior or the surface roughness. As a consequence, our results suggest that there is no change in the adhesion behavior of the material after numerous autoclaving cycles.

Regarding the dimensional changes, we reported a tendency of slight growth in the chosen geometry of the additive-manufactured stirrers along their autoclaving history. This is in accordance with the findings of Told et al., who reported an increase in cross-section by measurements with a caliper after 10 and 20 autoclaving cycles for a white SLA resin [10]. Studies by Marei et al. and Sharma et al. investigated the influence of autoclaving on materials for the printing of surgical guides, where high accuracy is required [15,16]. Marei et al. reported no significant changes in the center position of the integrated sleeves using mesh data generated by an inter-oral scanner [15]. In the study by Rynio et al., they observed similar results for aortic stent grafts printed by SLA and autoclaved once [13]. Sharma et al. detected, with a caliper, an increase in the outer dimensions of test parts too, but this was non-significant for SLA-manufactured specimens [16]. Studies by Keßler et al. and David et al., with intended applications in medical use, showed a dependency of deformation on the combination of resin material and vat-photopolymerization machine selected [17,18]. In an application study by Satzer et al., printed flasks made with a low-cost resin printer splintered after heating up to 120 °C and autoclaving afterwards [34]. Another study by Mahmud et al. investigated the degradation of SLA specimens manufactured with Clear V4 resin (Formlabs Inc., Somerville, MA, USA), subjected to deionized water over 12 weeks. They described swelling of the specimens through a mass increase, accompanied by a decrease in tensile strength, and explained this reduction with hydrolytic softening caused by the submersion in water [58]. Therefore, we assume that the decreased mechanical properties and the increased dimensions were, in part, caused by swelling due to water absorption while autoclaving. The ability of water absorption was also described by the manufacturer [37].

Considering the usage of additive-manufactured stirrers in bioprocess engineering, we detected no process-relevant deviations in the geometry by 3D scanning. This method of surface recording has an accuracy that ranges, depending on the regarded study, from 20 µm [59] up to 50 µm [60]. For applications where higher accuracy in geometry and surface measurement is required, more detailed investigations about the influence of autoclaving on dimensional changes would be necessary.

Most studies about the tightness of additive-manufactured parts were performed with leaking parts manufactured by material-extrusion processes and the optimization of printing parameters or post-processing for improved tightness [19,20,21,22]. Only a few studies have evaluated the tightness of parts produced by vat-photopolymerization-based additive manufacturing. Al-Hasni et al. reported tightness for SLA-manufactured parts by the air pressure or vacuum tests [23]. Andisetiawan et al. detected no significant leakage of electrolyte through SLA-printed parts [24]. In the study by Gensler et al., water-filled cups with walls of 2 mm lost nearly no fluid for 24 h [25]. These results are in line with our findings for the LCD–vat-photopolymerization-printed specimens, where we detected no air pressure leakage across twenty autoclaving cycles. To date, no other study we are aware of has evaluated the influence of autoclaving cycles on the tightness of comparable parts.

While this study focused on a single material and one sterilization method, it provides a valuable foundation for understanding the impact of autoclaving on AM components. The insights gained highlight the critical importance of material selection for maintaining properties over multiple sterilization cycles. When combined with findings from other studies, our results contribute to a growing body of knowledge emphasizing the need for heat-resistant materials. Further research can use the methods presented for other materials that are possibly suited for comparable applications. Potential investigations could be extended to other sterilization routines or the printing parameters. With a given bioprocess and microorganism, additional investigations addressing the cytotoxicity of the material or degradation products, as well as investigations regarding the biocontamination susceptibility, could be performed. Additional studies could also address the dimensional changes through autoclaving in more detail, especially regarding different geometries and larger sample sizes.

In conclusion, we state the suitability of xPeek147 for utilization in bioprocess applications inside the reported boundaries of our experimental results. This study provides a methodology for the assessment of whether an additive-manufactured material is suited or not for autoclaving over multiple cycles and, therefore, can be used outside of single-use applications in bioprocess engineering.

## 5. Conclusions

The aim of this study was to identify a material for LCD-based vat-photopolymerization additive manufacturing that is suited for steam autoclaving in multiple cycles. Therefore, a material described by the manufacturer as heat-resistant (xPeek147) was autoclaved and tested regarding its mechanical and surface properties, tightness behavior, and dimensional changes. Considering the limitations of this study, the following can be concluded from the reported results:For mechanical properties like tensile strength, elongation at break, and Charpy impact, decreasing values were reported first for specimens treated by 50 autoclaving cycles. For specimens treated with up to 20 autoclaving cycles, no significant changes in mechanical properties occurred.For surface properties, we detected no change in the surface roughness and no change in surface energies. The hydrophilic surface remained for all cycles.The additive-manufactured specimens for the tightness test stayed tight for the 20 autoclaving cycles tested.Regarding the dimensional changes in the tested benchmark bodies, no process-relevant deviations across ten autoclaving cycles were reported.

This study underlines the necessity of heat resistance in vat-photopolymerization materials that are intended to be autoclaved. Overall, we assign the investigated material a suitability for usage in bioprocess engineering in accordance with its properties. This allows the additive manufacturing of prototypes and functional components, as well as autoclaving in multiple cycles within the described limits.

## Figures and Tables

**Figure 1 materials-18-04720-f001:**
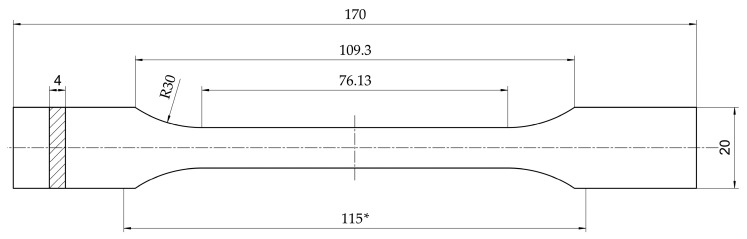
Dimensions of used tensile specimen. The dimension marked with * describes the distance between the clamping jaws (dimensions in mm). The formatting of the drawing is based on DIN EN ISO 527-2:2012-06 [39].

**Figure 2 materials-18-04720-f002:**
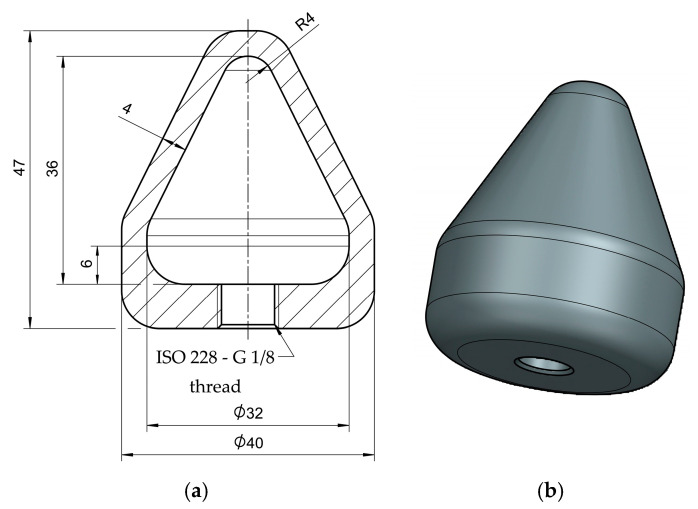
Dimensions (**a**) and three-dimensional view (**b**) of tightness testing body (all unspecified radii: 6 mm, dimensions in mm). The thread was designed according to DIN EN ISO 228-1:2003-05 [45].

**Figure 3 materials-18-04720-f003:**
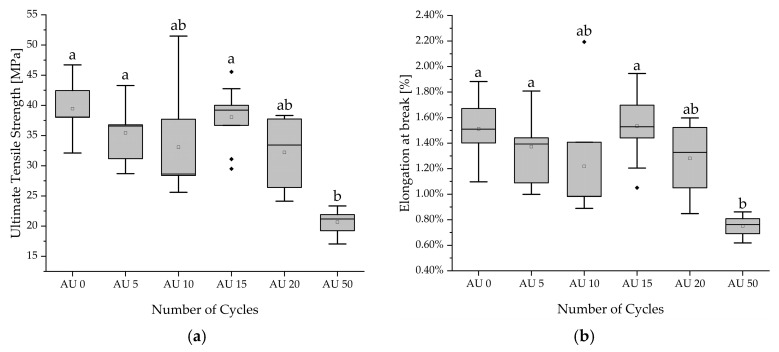
Boxplots of ultimate tensile strength (**a**) and elongation at break (**b**) for the different numbers of autoclaving cycles, respectively. Different alphabetical letters mark statistically significant differences. (▫) marks the mean value. Filled rhombic symbols represent the outliers.

**Figure 4 materials-18-04720-f004:**
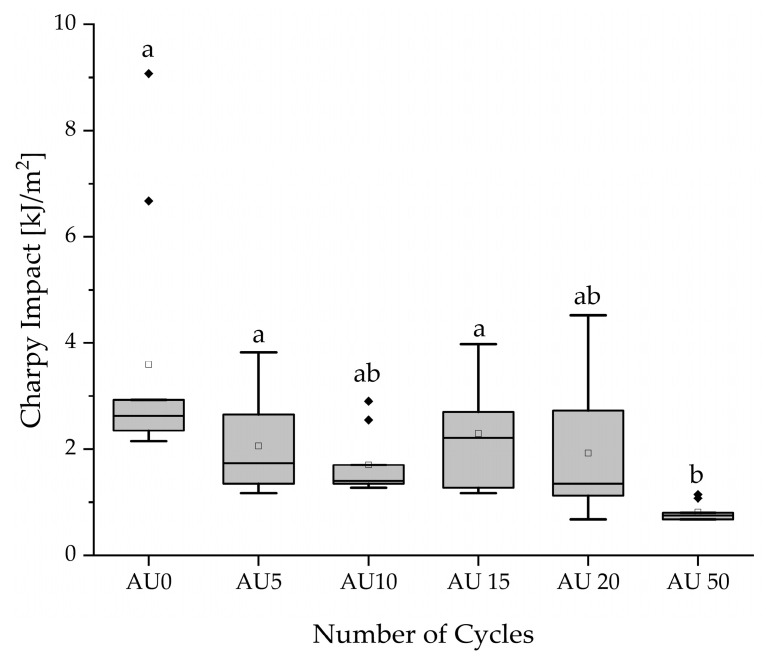
Boxplot: Results of Charpy impact across different autoclaving batches. Different alphabetical letters mark statistically significant differences. (▫) marks the mean value. Filled rhombic symbols represent the outliers.

**Figure 5 materials-18-04720-f005:**
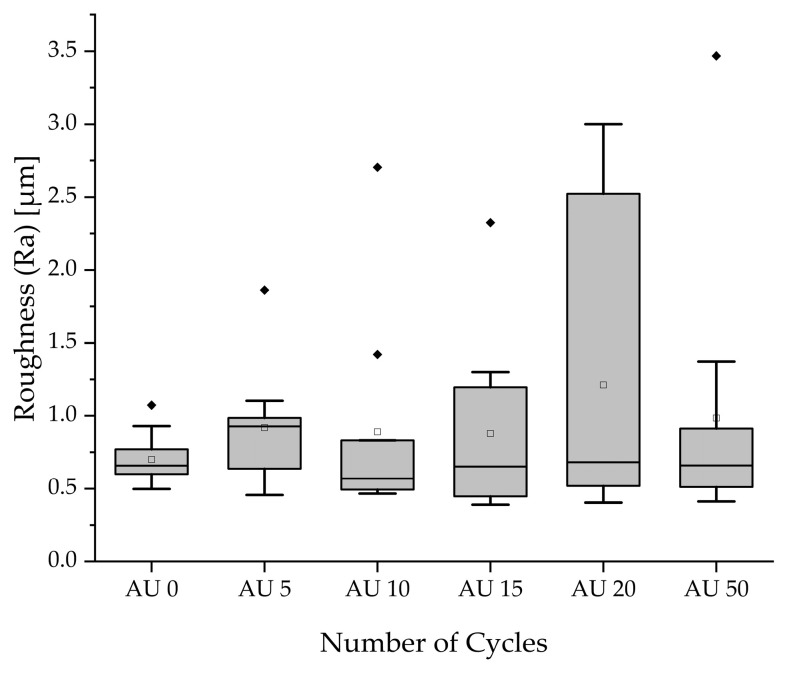
Boxplot: results of roughness measurements across different autoclaving batches with visualized maximum outliers responsible for large standard deviations. (▫) marks the mean value. Filled rhombic symbols represent the outliers.

**Figure 6 materials-18-04720-f006:**
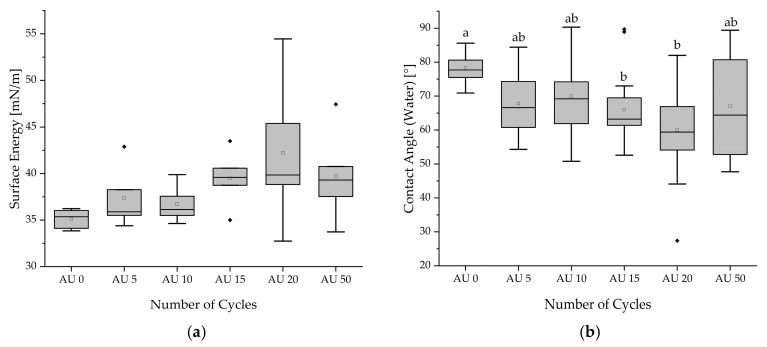
Boxplot: Values of surface energy (**a**) and contact angles for water (**b**) across different autoclaving batches. Different alphabetical letters mark statistically significant differences. (▫) marks the mean value. Filled rhombic symbols represent the outliers.

**Figure 7 materials-18-04720-f007:**
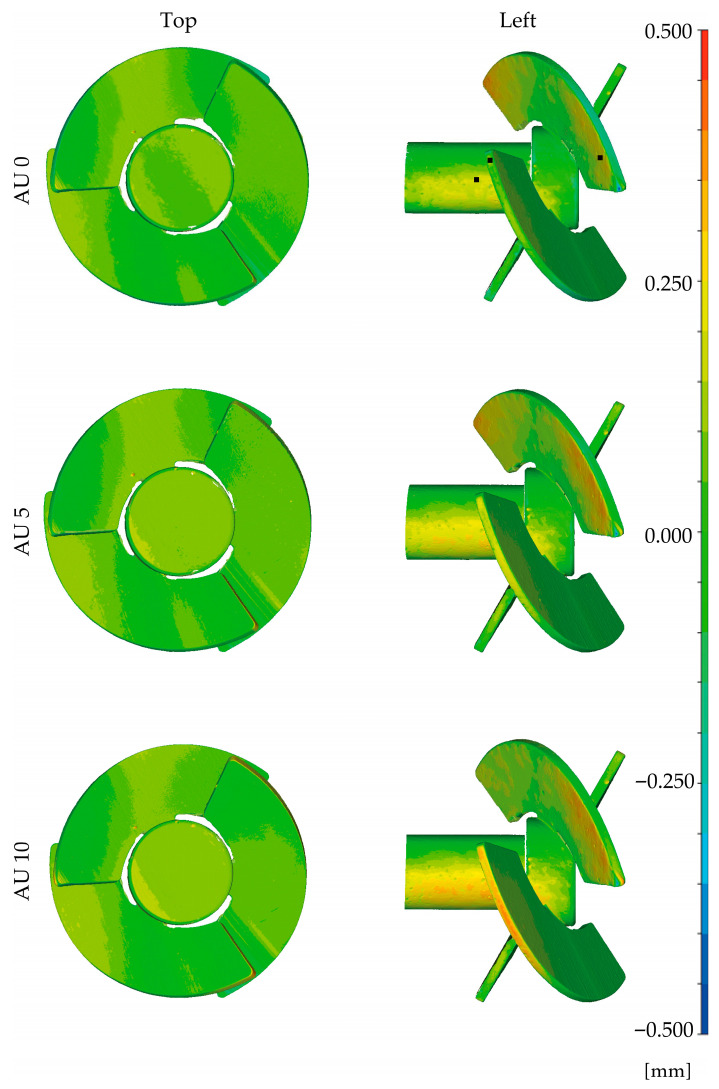
Results of the surface comparison of the first stirrer along the history of autoclaving batches: according to the scale, the surface coloring represents the 3D surface distances between CAD model and the recorded surface meshes. Investigated positions are marked with black squares.

**Table 1 materials-18-04720-t001:** The treatment of different specimen types.

		Autoclaving Cycles					
		0	5	10	15	20	50
**Specimen Type**	Tensile	×	×	×	×	×	×
	Charpy *	×	×	×	×	×	×
	Tightness **	×	×	×	×	×	
	Benchmark Bodies **	×	×	×			

* Charpy samples were also used for contact angle and roughness measurements; ** non-destructive tests and reuse of samples possible. (×) indicates. that the specimens were treated with the number of autoclaving cycles specified in the column.

**Table 2 materials-18-04720-t002:** Results of tensile testing for groups with different numbers of autoclaving cycles.

Group	Ultimate Tensile Strength [MPa] M ± SD	Bonferroni-Corrected Post Hoc Test **** *p*-Value	Elongation at Break [%] M ± SD	Bonferroni-Corrected Post Hoc Test **** *p*-Value
AU 0	39.47 ± 5.47	0.002 ** (AU0–AU50)	1.51 ± 0.29	0.008 ** (AU0–AU50)
AU 5	35.46 ± 5.04	0.020 * (AU5–AU50)	1.37 ± 0.29	0.024 * (AU5–AU50)
AU 10	33.07 ± 8.97	0.368 (AU10–AU50)	1.22 ± 0.46	-
AU 15	38.04 ± 5.11	<0.001 *** (AU15–AU50)	1.53 ± 0.28	<0.001 *** (AU15–AU50)
AU 20	32.20 ± 5.93	0.287 (AU20–AU 50)	1.28 ± 0.28	0.082 (AU20–AU50)
AU 50	20.65 ± 2.24	-	0.85 ± 0.05	-

* *p* < 0.05; ** *p* < 0.01; *** *p* < 0.001; **** filtered results with *p* < 0.5 with test comparison, respectively.

**Table 3 materials-18-04720-t003:** Results of Charpy impact test and statistical analysis for significant differences.

Group	Charpy Impact [kJ/m^2^] M ± SD	Bonferroni-Corrected Post Hoc Test ****
AU 0	3.60 ± 2.34	<0.001 *** (AU0–AU50)
AU 5	2.06 ± 0.88	0.014 * (AU5–AU50)
AU 10	1.70 ± 0.60	0.192 (AU10–AU50)
AU 15	2.29 ± 1.02	0.005 ** (AU15–AU50)
AU 20	1.93 ± 1.37	0.157 (AU20–AU50)
AU 50	0.81 ± 0.18	-

* *p* < 0.05; ** *p* < 0.01; *** *p* < 0.001; **** filtered results with *p* < 0.2 with test comparison, respectively.

**Table 4 materials-18-04720-t004:** Results of roughness measurements.

Group	Roughness R_a_ * [µm] M ± SD
AU 0	0.70 ± 0.18
AU 5	0.92 ± 0.39
AU 10	0.89 ± 0.70
AU 15	0.88 ± 0.61
AU 20	1.21 ± 1.03
AU 50	0.98 ± 0.92

* R_a_: arithmetically averaged roughness.

**Table 5 materials-18-04720-t005:** Results of surface energy determination and contact angles with statistical evaluation.

Group	Surface Energy [mN/m] M ± SD	Contact Angle Water [°] M ± SD	Bonferroni-Corrected Post Hoc Test **** (*p*-Value)
AU 0	35.12 ± 1.09	78.3 ± 3.9	-
AU 5	37.39 ± 3.38	67.8 ± 8.5	0.089
AU 10	36.75 ± 2.05	70.0 ± 10.1	0.380
AU 15	39.48 ± 3.08	66.0 ± 10.7	0.016 *
AU 20	42.25 ± 8.16	60.0 ± 14.3	<0.001 ***
AU 50	39.75 ± 5.03	67.1 ± 14.4	0.073

* *p* < 0.05; *** *p* < 0.001; **** batch compared with reference AU0; results called significant for *p* < 0.05.

## Data Availability

The original data presented in this study are openly available in ZENODO at DOI: 10.5281/zenodo.15275242.

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
