# Peer review of "Impact of Autoclaving on the Material Properties of Vat-Photopolymerization-Produced Components Intended for Bioprocess Engineering"

_materials, 2025, doi:10.3390/ma18204720_

Round 1
Reviewer 1 Report (Previous Reviewer 1)
Comments and Suggestions for Authors
This article is quite interesting and addresses a relevant topic related to the study of materials for the additive manufacturing of repeatedly sterilizable components in biotechnology. The integrated approach is noteworthy: the authors examined mechanical and surface properties and used a large number of autoclaving cycles. However, a number of questions and comments arise regarding the work:
1) The first concern concerns the measurement of dimensional changes. Two samples is certainly too small a sample and cannot yield statistically significant conclusions.
2) The authors need to discuss in more detail the role of hydrolysis and thermal-oxidative degradation in the context of microcrack formation after 50 cycles and the subsequent decline in mechanical properties.
3) The authors cite water absorption as the cause of the material's size increase, but do not measure it. This needs to be addressed.
4) The dependence of degradation on sample geometry should also be studied to identify the influence of both volume and surface area.
5) Finally, the work needs to be supplemented by a study of resistance to biocontamination, one of the key factors for bioengineering.
Author Response
Please see the attachment.

Reviewer 2 Report (Previous Reviewer 2)
Comments and Suggestions for Authors
Publish as it is.
Author Response
Please see the attachment.

Reviewer 3 Report (Previous Reviewer 3)
Comments and Suggestions for Authors
In this work, the authors studied a material for LCD-based vat-photopolymerization additive manufacturing. This work is rich in data and clear in discussion, and the results are meaningful. In view of this, this work can be accepted after minor revisions.
The error markings on the image are not very clear, please ask the author to modify them.
Please discuss in detail the research progress of relevant work in recent years.
The content of the preface can be appropriately shortened.
Round 2
Reviewer 1 Report (Previous Reviewer 1)
Comments and Suggestions for Authors
The authors responded to all comments and made the necessary adjustments. The article may be published in its current form.
This manuscript is a resubmission of an earlier submission. The following is a list of the peer review reports and author responses from that submission.
Round 1
Reviewer 1 Report
Comments and Suggestions for Authors
The presented article concerns the study of the effect of autoclaving on the properties of compounds obtained by photopolymerization. This is a fairly relevant area, expanding approaches to the design of materials with specified structure and properties. The authors demonstrated a comprehensive approach to assessing mechanical properties and surface studies. An important advantage is the assessment of practical significance for medical applications. At the same time, there are a number of questions and comments to the authors:
1) The abstract should reduce the introductory part and add more information that constitutes the essence of the study. The authors did not even indicate which materials were studied.
2) The experimental part should include information on the heating rate of the material. This can be very important for the reproducibility of the results.
3) How was the possibility of oxidation of the samples assessed? Have the authors tried to use IR spectroscopy for this?
4) To what extent did the authors exclude the possibility of polymer hydrolysis? This can significantly affect the strength of the material. In connection with the likelihood of hydrolysis, it is appropriate to indicate the humidity level in the autoclave.
5) It is necessary to examine the surface of the samples after autoclaving for microcracks, for example, using SEM microscopy.
6) It is necessary to assess the toxicity of the polymer degradation products (since we are talking about medical use)
Reviewer 2 Report
Comments and Suggestions for Authors
The author use the table and the passive data to study the Material Properties, which is queer.
Reviewer 3 Report
Comments and Suggestions for Authors
In this work, the authors evaluate the influence of multiple autoclaving cycles on the properties of a material printed with vat photopoly-merization. The work seems meaningful, and the discussion of the results is also thorough. However, there are still some issues that need to be addressed by the author. Specifically, as follows:
- The current research progress is not comprehensive. Please supplement the research progress related to this work in the
- The author cites necessary references in section 3.6 Testing for dimensional changes
- In Table 4, please add a footnote to Ra by the author.
- In the abstract section, please ask the author to provide more data descriptions to highlight the highlights of this work more intuitively.
